# Investigating the possible causal association of smoking with depression and anxiety using Mendelian randomisation meta-analysis: the CARTA consortium

Amy E Taylor,[1,2] Meg E Fluharty,[1,2] Johan H Bjørngaard,[3,4]
Maiken Elvestad Gabrielsen,[5] Frank Skorpen,[5] Riccardo E Marioni,[6,7,8]
Archie Campbell,[7] Jorgen Engmann,[9] Saira Saeed Mirza,[10] Anu Loukola,[11]
Tiina Laatikainen,[12,13,14] Timo Partonen,[15] Marika Kaakinen,[16,17] Francesca Ducci,[18]
Alana Cavadino,[19] Lise Lotte N Husemoen,[20] Tarunveer Singh Ahluwalia,[21,22,23]
Rikke Kart Jacobsen,[20] Tea Skaaby,[20] Jeanette Frost Ebstrup,[20]
Erik Lykke Mortensen,[24] Camelia C Minica,[25] Jacqueline M Vink,[25]
Gonneke Willemsen,[25] Pedro Marques-Vidal,[26] Caroline E Dale,[27] Antoinette Amuzu,[27]
Lucy T Lennon,[28] Jari Lahti,[29,30] Aarno Palotie,[31,32,33] Katri Räikkönen,[30]
Andrew Wong,[34] Lavinia Paternoster,[1,35] Angelita Pui-Yee Wong,[36,37]
L John Horwood,[38] Michael Murphy,[39] Elaine C Johnstone,[40] Martin A Kennedy,[41]
Zdenka Pausova,[42,43] Tomáš Paus,[37,44] Yoav Ben-Shlomo,[35] Ellen A Nohr,[45]
Diana Kuh,[34] Mika Kivimaki,[9] Johan G Eriksson,[30,46,47,48,49,50] Richard W Morris,[28]
Juan P Casas,[27,51] Martin Preisig,[52] Dorret I Boomsma,[25] Allan Linneberg,[20,53,54]
Chris Power,[19] Elina Hyppönen,[19,55,56] Juha Veijola,[57] Marjo-Riitta Jarvelin,[16,17,58,59,60]
Tellervo Korhonen,[11,12,15] Henning Tiemeier,[61] Meena Kumari,[9] David J Porteous,[6,7]
Caroline Hayward,[62] Pål R Romundstad,[3] George Davey Smith,[1,35]
Marcus R Munafò[1,2]

▶ Prepublication history and additional material is available. To view please visit the journal (http://dx.doi.org/10.1136/bmjopen-2014-006141).

For numbered affiliations see end of article.

**Correspondence to**
Dr Amy Taylor;
amy.taylor@bristol.ac.uk

## ABSTRACT

**Objectives:** To investigate whether associations of smoking with depression and anxiety are likely to be causal, using a Mendelian randomisation approach.

**Design:** Mendelian randomisation meta-analyses using a genetic variant (rs16969968/rs1051730) as a proxy for smoking heaviness, and observational meta-analyses of the associations of smoking status and smoking heaviness with depression, anxiety and psychological distress.

**Participants:** Current, former and never smokers of European ancestry aged ≥16 years from 25 studies in the Consortium for Causal Analysis Research in Tobacco and Alcohol (CARTA).

**Primary outcome measures:** Binary definitions of depression, anxiety and psychological distress assessed by clinical interview, symptom scales or self-reported recall of clinician diagnosis.

**Results:** The analytic sample included up to 58 176 never smokers, 37 428 former smokers and 32 028 current smokers (total N=127 632). In observational analyses, current smokers had 1.85 times greater odds of depression (95% CI 1.65 to 2.07), 1.71 times greater odds of anxiety (95% CI 1.54 to 1.90) and 1.69 times greater odds of psychological distress (95% CI 1.56 to 1.83) than never smokers. Former smokers also had greater odds of depression, anxiety and psychological distress than never smokers. There was evidence for

## Strengths and limitations of this study

- This is the largest Mendelian randomisation study of the relationship between smoking and depression and anxiety conducted to date.
- By using a genetic variant associated with smoking heaviness as a proxy for smoking heaviness, bias from confounding is minimised and findings not affected by reverse causality.
- Measurement of depression and anxiety differed across studies so we were unable to use a consistent definition.
- While results are consistent with no causal association between smoking heaviness and depression or anxiety, we cannot rule out the possibility of a small effect.

positive associations of smoking heaviness with depression, anxiety and psychological distress (ORs per cigarette per day: 1.03 (95% CI 1.02 to 1.04), 1.03 (95% CI 1.02 to 1.04) and 1.02 (95% CI 1.02 to 1.03) respectively). In Mendelian randomisation analyses, there was no strong evidence that the minor allele of rs16969968/rs1051730 was associated with depression (OR=1.00, 95% CI 0.95 to 1.05), anxiety (OR=1.02, 95% CI 0.97 to 1.07) or psychological distress (OR=1.02, 95% CI 0.98 to 1.06) in current smokers. Results were similar for former smokers.

**Conclusions:** Findings from Mendelian randomisation analyses do not support a causal role of smoking heaviness in the development of depression and anxiety.

## INTRODUCTION

Smoking is highly comorbid with both depression and anxiety across many different populations.[1–9] Furthermore, there is evidence to suggest that tobacco control interventions may not be as effective in populations with mental health conditions; for example, recent trends in the USA suggest that, since 2004, smoking rates have declined less rapidly in individuals with anxiety than in the general population.[10] Given the profound public health burden of both tobacco-related disease,[11] and depression and anxiety,[12] understanding this relationship is of great importance.

Unfortunately, it is difficult to infer causal links between smoking and depression and anxiety from observational data, due to confounding. There may be factors associated with both smoking and depression and anxiety, such as other substance use (eg, alcohol), socioeconomic adversity and education which cannot be fully accounted for in observational studies.[13] In addition, even if a causal association does exist, the direction of the relationship between smoking and depression and anxiety is unclear.[14] Prospective studies have provided evidence that depressive symptoms are associated with increased likelihood of smoking initiation,[2 7 15–17] while smoking cessation appears to be associated with a short-term increase in depressive symptoms during their quit attempt among a subgroup of smokers, and these individuals have poor smoking cessation outcomes.[18] This evidence is consistent with the popular belief that cigarette smoking can reduce anxiety and improve mood, particularly among those experiencing anxiety or low mood (the self-medication hypothesis). However, there is also a growing body of evidence suggesting that smoking may contribute to the development of these conditions[2 7 19–21] and that smoking cessation is associated with improvements in mental health, including depression and anxiety, compared to continued smoking.[22]

Plausible biological mechanisms through which constituents of tobacco smoke may cause depression and anxiety have been described. In animal studies, for example, there is evidence that nicotine administration produces dysregulation in the hypothalamic-pituitary-adrenal system, which leads to hypersecretion of cortisol and changes in the activity of associated monoamine neurotransmitters.[23] These systems function to regulate the biological and psychological reactions to stressors. Similarly, human data have demonstrated elevated cortisol levels in smokers compared to non-smokers.[24] Constituents of tobacco smoke inhibit the activity of monoamine oxidase, enzymes that are involved in the breakdown of monoamines (including dopamine, serotonin and norepinephrine); this effect appears to normalise following smoking cessation.[25] Animal studies also indicate that both drugs of abuse (including nicotine) and environmental stressors appear to trigger changes in midbrain dopaminergic function.[26] Consequently, prolonged smoking may act to sensitise stress response systems, weakening adaptive coping responses and making smokers more susceptible to emotional distress in response to environmental stressors.

Mendelian randomisation methods allow us to investigate causal relationships in humans by using genetic variants as proxies for exposures of interest. The principle of Mendelian randomisation relies on the basic (but approximate) laws of Mendelian genetics (segregation and independent assortment). When these principles hold, genetic variants, at a population level, will not be associated with the confounding factors that generally distort conventional observational studies.[27 28] In addition, as an outcome measure cannot alter the genotype that an individual is born with, these analyses should not be biased by reverse causality. A genetic variant, single nucleotide polymorphism (SNP) number rs16969968, in the *CHRNA5-CHRNA3-CHRNB4* (*CHRNA5-A3-B4*) nicotinic receptor subunit gene cluster on chromosome 15 has demonstrated robust association with smoking heaviness within smokers.[29–32] The rs16969968 variant is functional and leads to an amino acid change (D398N) in the nicotinic receptor α5 subunit protein.[33] The minor (risk) allele of this variant is associated with an average increase in smoking amount of one cigarette per day in smokers, and even more strongly associated with increases in cotinine (a metabolite of nicotine) levels.[31 34 35] However, given the known role of the variant in altering receptor function,[33] it is likely that the greater variance explained for cotinine levels is due to this measure better capturing total tobacco exposure, and not because the variant directly affects nicotine metabolism.[31] There is also good evidence that the rs16969968 variant, unlike smoking heaviness, does not associate with confounding factors that may distort associations with health outcomes, for example, socioeconomic status and education level.[36 37]

The rs16969968 variant (or its proxy rs1051730) has been used as an instrument for smoking heaviness in Mendelian randomisation studies to demonstrate that smoking causally lowers body mass index[38] and that maternal smoking during pregnancy lowers offspring birth weight[39] (see figure 1 for an illustration of the Mendelian randomisation approach). Using the rs1051730 variant, two recent studies have applied Mendelian randomisation to examine the causal

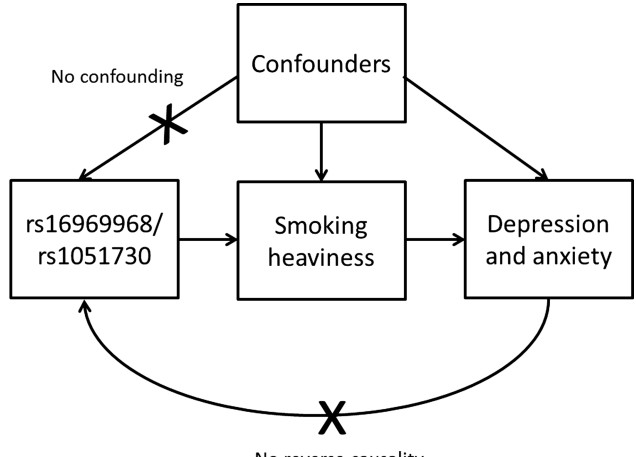

**Figure 1** Diagram of Mendelian randomisation analysis of smoking and depression/anxiety. The genetic variant rs16969968/rs1051730 is associated with smoking heaviness but should not be associated with the confounders of the association between smoking heaviness and depression/anxiety. In addition, there is no pathway from depression and anxiety to the genetic variant (reverse causality).

relationship of smoking with depression and anxiety.[37 38] In one large Norwegian population, the rs1051730 variant was not associated with depression or anxiety in smokers[40]; in a British cohort, the rs1051730 variant was associated with decreased depression during pregnancy in women who smoked prior to pregnancy.[41] These findings are not consistent with a causal role of smoking in increasing depression or anxiety. To test the robustness of these findings, we performed a Mendelian randomisation meta-analysis combining data from 25 studies (n=127 632) in the consortium for Causal Analysis Research in Tobacco and Alcohol (CARTA).

## METHODS
### Study populations
We used data on individuals aged ≥16 years and of self-reported European ancestry from 25 studies from the CARTA consortium: the 1958 Birth Cohort (1958BC), the Avon Longitudinal Study of Parents and Children (ALSPAC, including both mothers and children), the British Regional Heart Study (BRHS), the British Women's Heart and Health Study (BWHHS), the Caerphilly Prospective Study (CaPS), the Christchurch Health and Development Study (CHDS), Cohorte Lausannoise (CoLaus), the English Longitudinal Study of Ageing (ELSA), the National FINRISK Study (FINRISK), Generation Scotland: the Scottish Family Health Study (GS:SFHS), Genomics of Overweight Young Adults (GOYA) females, the Helsinki Birth Cohort Study (HBCS), Health2006, Health2008, the second wave of the Nord-Trøndelag health study (HUNT 2), Inter99, the Northern Finland Birth Cohorts (NFBC1966 and NFBC1986), the National Health and Nutrition Examination Survey (NHANES), the MRC National

Survey of Health and Development (NSHD), the Netherlands Twin Registry (NTR), Patch 2, the Rotterdam Study, the Saguenay Youth Study-Parents (SYS-P) and the Whitehall II study. Further details of these studies are provided in online supplementary material.

### Genotype
Within each study, individuals were genotyped for one of two SNPs in the *CHRNA5-A3-B4* nicotinic receptor subunit gene cluster, rs16969968 or rs1051730. These SNPs are in perfect linkage disequilibrium with each other in Europeans ($R^2$=1.00 in HapMap 3, http://hapmap.ncbi.nlm.nih.gov/) and therefore represent the same genetic signal. Where studies had data available for both SNPs, we used the SNP that was genotyped in the largest number of individuals. Details of genotyping methods within each study are provided in online supplementary material.

### Measures of depression, anxiety and psychological distress
Depression and anxiety were assessed by clinical interview, symptom scales or self-reported recall of clinician diagnosis (see table 1). As some of the scales do not distinguish between symptoms of depression and anxiety, we used the term 'psychological distress' to refer a composite phenotype.

To compare measures across studies, we created two case definitions for each of depression, anxiety and psychological distress (see table 2). According to case definition 1, individuals were classified as depressed or anxious if they self-reported clinician diagnosis of depression or anxiety, met clinical criteria for depression (excluding bereavement where known) or anxiety, or were above previously published cut points for depression or anxiety on symptom scales. Individuals were classified as having psychological distress if they met case definition 1 for depression or anxiety, or if they were above a cut point on a general scale for psychiatric symptoms. As not all scale measures used for assessing mental health have published cut-offs for defining cases, we created a second definition. According to case definition 2, individuals were classified as depressed or anxious if they were above the 90th centile for the specific depression or anxiety scales, and psychologically distressed if they were above the 90th centile on *either* the depression or anxiety scales or above the 90th centile on the general scales of psychiatric symptoms. Where both case definitions 1 and 2 were available within a study, case definition 1 was used. Full details of the measures and cut points used are provided in online supplementary table S1.

For the majority of studies (k=17), diagnoses were based on current depression and anxiety (at the time of measurement). Where current diagnoses were not available, diagnoses of depression or anxiety in the previous 12 months or lifetime diagnoses were used. For lifetime diagnoses, if information on age at first diagnosis was collected, individuals reporting diagnoses prior to 16 years of age were excluded.

**Table 1** Measures of depression, anxiety and psychological distress in the CARTA studies

| Study | Psychological Distress | Depression | Anxiety |
|---|---|---|---|
| 1958BC | CIS-R | CIS-R | CIS-R |
| ALSPAC children | CIS-R | CIS-R | CIS-R |
| ALSPAC mothers | EPDS or CCEI | EPDS | CCEI |
| BRHS | | Clinician diagnosis (lifetime) | |
| BWHHS | | Clinician diagnosis (lifetime) | |
| CaPS | GHQ-30 | | STAI |
| CHDS | CIDI (previous 12 months) | CIDI (previous 12 months) | CIDI (previous 12 months) |
| CoLaus | DIGS | DIGS | DIGS |
| ELSA | CES-D or clinician diagnosis of anxiety | CES-D (8-item) | Clinician diagnosis (lifetime) |
| FINRISK | | Clinician diagnosis (previous 12 months) | |
| Generation Scotland | GHQ-28 | SCIDI/NP (Lifetime diagnosis) | |
| GOYA females | Clinician diagnosis (since giving birth) | Clinician diagnosis (since giving birth) | Clinician diagnosis (since giving birth) |
| HBCS | CES-D or STAI | CES-D (20 items) | STAI |
| Health2006 | SCL-90-R | SCL-90-R | SCL-90-R |
| Health2008 | SCL-90-R | SCL-90-R | SCL-90-R |
| HUNT | HADS | HADS | HADS |
| Inter99 | SCL-90-R | SCL-90-R | SCL-90-R |
| NFBC1966 | SCL-25 | SCL-25 | SCL-25 |
| NFBC1986 | YSR | YSR | YSR |
| NHANES | | DIS (lifetime diagnosis) | |
| NSHD | GHQ-28 | | |
| NTR | ASR | ASR | ASR |
| Patch 2 | | SCID (lifetime diagnosis) | |
| Rotterdam | CES-D or M-CIDI | CES-D (20 items) | M-CIDI |
| SYS-P | CES-D or DSM instrument | CES-D (12 items) | 10 question DSM-based instrument |
| Whitehall II | GHQ-30 | | |

All scales measure current depression and anxiety unless otherwise stated. Clinician diagnosis was assessed by self-reported recall in all studies.
ALSPAC, Avon Longitudinal Study of Parents and Children; ASR, Adult Self Report; BC, Birth Cohort; BRHS, British Regional Heart Study; BWHHS, British Women's Heart and Health Study; CARTA, Causal Analysis Research in Tobacco and Alcohol; CCEI, Crown–Crisp Experiential Index; CES-D, Centre for Epidemiologic Studies Depression; CHDS, Christchurch Health and Development Study; CIDI, Composite International Diagnostic Interview; CIS-R, Computerised interview schedule-revised; CoLaus, Cohorte Lausannoise; DIGS, Diagnostic Interview for Genetic Studies; DIS, Diagnostic Interview Schedule; ELSA, English Longitudinal Study of Ageing; EPDS, Edinburgh Postnatal Depression Scale; GHQ, General Health Questionnaire; GOYA, Genomics of Overweight Young Adults; HADS, Hospital Anxiety and Depression Scale; HBCS, Helsinki Birth Cohort Study M-CIDI, Munich version of CIDI; NHANES, National Health and Nutrition Examination Survey; NSHD, National Survey of Health and Development; NTR, the Netherlands Twin Registry; SCID, Structured Clinical Interview for DSM-III-R diagnosis; SCIDI/NP, Structured Clinical Interview for DSM-IV Axis disorders non-patient edition; SCL, symptoms checklist; STAI, State Trait Anxiety Inventory; YSR, Youth Self Report.

Symptom scales were also used as continuous measures of depression, anxiety and psychological distress (see online supplementary table S2). To compare across studies, these were converted to z-scores within each study. Most measures of depression, anxiety and psychological distress were strongly right skewed. However, standard transformations (eg, log, square root) did not greatly improve distributions in several of the samples. Therefore, z-scores were constructed using the untransformed data in all samples (z-score=(individual score −sample mean)/sample SD).

**Smoking status**

Smoking status was self-reported (either by questionnaire or interview) at the same time as mental health assessment for all studies, with the exception of 1958BC, CoLaus and HBCS which used smoking status and depression/anxiety data collected up to 3 years apart (see online supplementary material). Individuals were classified as never, former, current or ever (ie, current and former combined) cigarette smokers. Where information on smoking frequency was available, current smokers were restricted to individuals smoking at least one cigarette per day. Where information on pipe and cigar smoking was available, individuals reporting being current or former smokers of pipes or cigars but not cigarettes were excluded from all analyses.

For studies with adolescent populations (ALSPAC children and NFBC1986), analyses were restricted to current daily smokers who reported smoking at least one

**Table 2** Case definitions for depression, anxiety and psychological distress

|  | Case definition 1 | Case definition 2 |
|---|---|---|
| Depression | Self-report of clinical diagnosis **OR** Meeting clinical criteria for depression **OR** Scoring above published cut point on specific depression scale | Scoring >90th centile on specific depression scale |
| Anxiety | Self-report of clinical diagnosis **OR** Meeting clinical criteria for anxiety disorders **OR** Scoring above published cut point on specific anxiety scale | Scoring >90th centile on specific anxiety scale |
| Psychological distress | Depression or anxiety as defined above **OR** Scoring above published cut point on general scale for psychiatric symptoms | Depression or anxiety as defined above **OR** >90th centile on general scale for psychiatric symptoms |

cigarette per day (current smokers) and individuals who had never tried smoking (never smokers).

Smoking heaviness in current smokers, measured as cigarettes smoked per day, was collected in some studies as a continuous variable and in some studies as a categorical variable. Further details of the smoking measures collected within each study are provided in the online supplementary material.

## Statistical analysis

Analyses were conducted within each contributing study using Stata or R software, following the same analysis plan. Analyses were restricted to individuals with full data on depression and anxiety outcomes, smoking status and rs16969968/rs1051730 genotype.

Sex-adjusted and age-adjusted associations of smoking status (never, former, current, ever) and smoking heaviness with binary measures of depression, anxiety and psychological distress were assessed using logistic regression. For the smoking status analysis, never smokers were used as the reference group. The smoking heaviness analysis was restricted to current daily smokers, and ORs represent differences in odds of the outcome measure per additional cigarette consumed per day. These analyses were restricted to studies with continuous measures of cigarettes per day.

Within each study, genotype frequencies were tested for deviation from Hardy-Weinberg equilibrium (HWE) using a $\chi^2$ exact test. Mendelian randomisation analyses of the association between rs16969968/rs1051730 and binary measures of depression, anxiety and psychological distress were performed using logistic regression and adjusted for age and sex. Analyses were performed stratified by smoking status (never, former, current and ever), because the variant only influences smoking heaviness in individuals who smoke. The analysis in never smokers is a test of a key assumption of Mendelian

randomisation: that the gene only operates on the outcome through its effects on smoking heaviness (ie, no pleiotropy). If rs16969968/rs1051730 only operates on an outcome measure through smoking heaviness, no association should be observed in never smokers. An additive genetic model was assumed, so ORs represent the difference in odds of the outcome per additional copy of the minor (risk) allele. As a secondary analysis, Mendelian randomisation analyses were performed of the association of rs16969968/rs1051730 with z-scores of symptoms scales for depression, anxiety and psychological distress using linear regression stratified by smoking status. These analyses were adjusted for age and sex and additionally for use of depression or anxiety medication where available. For studies with a survey (NHANES) or family-based design (SYS-P), appropriate methods were used to adjust SEs (see online supplementary material for further information). ALSPAC mothers and children were analysed as separate samples but, as mothers and children were related, sensitivity analyses were performed excluding each one of these samples.

Results from individual studies were meta-analysed in Stata (V.11) using the 'metan' command. Where there was evidence of heterogeneity between studies ($I^2 > 50\%$), both fixed and random effects analyses were performed. Within the Mendelian randomisation analyses, the Cochran Q statistic was used to test for interactions between genotype and smoking status on the outcome measures.

Analyses were also performed stratified by sex because there is some evidence from observational studies that the association between smoking and mental health outcomes may differ by sex.[42 43] Sex differences in the association between genotype and outcomes measures were tested for using meta-regression after taking into account potential differences by smoking status.

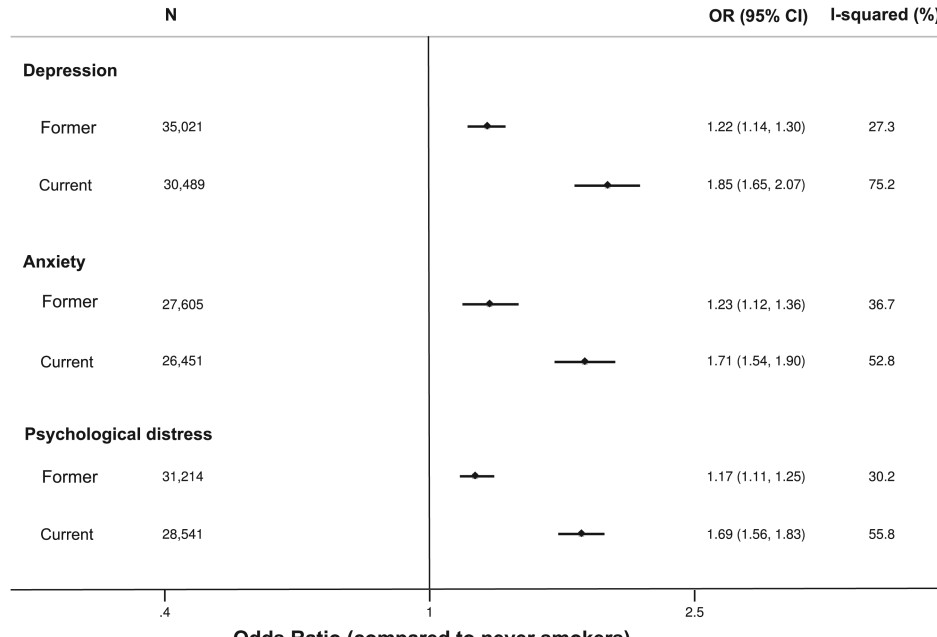

**Figure 2** Age-adjusted and sex-adjusted association of smoking status with depression, anxiety and psychological distress.

## RESULTS

### Descriptive statistics

In total, data on up to 127 632 individuals were available for analysis, including 58 176 never smokers, 37 428 former smokers and 32 028 current smokers. Overall, 45% of the combined study population was male. The median age within the contributing studies ranged between 16 and 68 years. The mean prevalence of depression, anxiety and psychological distress (using case definition 1) was 12.5% (range 6.1–37.5%), 10.2% (range 2.6–19.9%) and 17.4% (range 8.9–27%), respectively. Descriptive statistics for each of the study populations are provided in the online supplementary table S3.

The minor allele frequency for rs16969968/rs1051730 ranged between 0.31 and 0.39 (see online supplementary table S4). There was no strong evidence for deviation from HWE in any of the studies (p values all ≥0.06).

### Observational analysis

Levels of depression, anxiety and psychological distress differed by smoking status (see figure 2). In age-adjusted and sex-adjusted analyses, current smokers had 1.85 times (95% CI 1.65 to 2.07, p<0.001) greater odds of depression, 1.71 times (95% CI 1.54 to 1.90, p<0.001) greater odds of anxiety and 1.69 times (95% CI 1.56 to 1.83, p<0.001) greater odds of psychological distress than never smokers. Former smokers had 1.22 times (95% CI 1.14 to 1.30, p<0.001) greater odds of depression, 1.23 times (95% CI 1.12 to 1.36, p<0.001) greater odds of anxiety and 1.17 times (95% CI 1.11 to 1.25, p<0.001) greater odds of psychological distress than never smokers.

Among smokers, smoking heaviness was positively associated with levels of depression, anxiety and psychological distress (figure 3). In age-adjusted and sex-adjusted analyses, a one cigarette per day increase in smoking heaviness was associated with a 1.03-fold (95% CI 1.02 to 1.04, p<0.001) increase in the odds of having depression, a 1.03-fold (95% CI 1.02 to 1.04, p<0.001) increase in the odds of having anxiety and a 1.02-fold (95% CI 1.02 to 1.03, p<0.001) increase in the odds of having psychological distress.

As there was evidence of between-study heterogeneity for analyses of both smoking status and smoking heaviness, random effects meta-analyses are presented. However, results from fixed effects meta-analyses were similar (data not shown). Individual study estimates for observational analyses are provided in online supplementary figures S1 and S2.

### Mendelian randomisation analysis

There was no clear evidence that rs16969968/rs1051730 was associated with binary measures of depression in never (OR per minor allele 1.02, 95% CI 0.97 to 1.06, p=0.47), former (OR 1.00, 95% CI 0.95 to 1.05, p=0.99), current (OR 1.00, 95% CI 0.95 to 1.05, p>0.99) or ever (OR 1.01, 95% CI 0.98 to 1.05, p=0.58) smokers (see figure 3). Similarly, there was no clear evidence that rs16969968/rs1051730 was associated with binary measures of anxiety in former (OR 1.02, 95% CI 0.97 to 1.08, p=0.44), current (OR 1.02, 95% CI 0.97 to 1.07, p=0.42) or ever (OR 1.03, 95% CI 0.99 to 1.07, p=0.16) smokers. However, in never smokers there was some evidence of a positive association between the minor allele of rs16969968/rs1051730 and anxiety (OR 1.05, 95% CI 1.01 to 1.10, p=0.03). For psychological distress there was a similar pattern, with no strong evidence for an association between rs16969968/rs1051730 in smokers, but some evidence of a positive association in never smokers

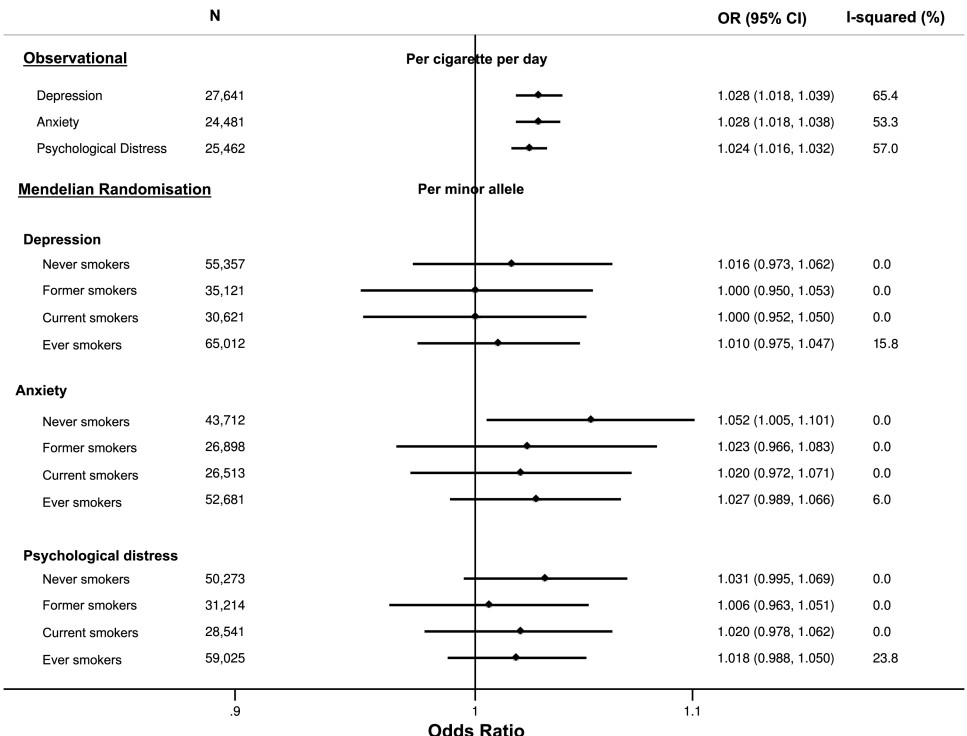

| | N | Per cigarette per day | OR (95% CI) | I-squared (%) |
|---|---|---|---|---|

**Figure 3** Age-adjusted and sex-adjusted observational and Mendelian randomisation analyses of association of smoking heaviness with depression, anxiety and psychological distress. Observational analysis performed using random effects meta-analysis and Mendelian randomisation analysis performed using fixed effects meta-analysis. Observational analysis restricted to current smokers.

(OR 1.03, 95% CI 1.00 to 1.07, p=0.09). For all outcomes, there was no clear statistical evidence for a difference in the effect of rs16969968/rs1051730 between never, former and current smokers (p values for heterogeneity between never, former and current smokers from Cochran Q test all >0.57). Individual study estimates for observational analyses are provided in online supplementary figure S3.

Results were similar for continuous measures of symptoms of depression, anxiety and psychological distress (see online supplementary figures S4 and S5). There was no clear evidence for associations of rs16969968/rs1051730 with continuous outcomes in any of the smoking categories.

Finally, there was no clear evidence for sex differences in either observational or Mendelian randomisation analyses for associations between smoking or smoking-related genotype and depression or anxiety (data available on request).

## DISCUSSION
In the largest Mendelian randomisation study on the association of smoking with depression and anxiety conducted to date, we found no evidence to suggest that smoking causes either depression or anxiety. Despite higher levels of depression, anxiety and psychological distress in current and former smokers compared to never smokers, and a positive association between the

number of cigarettes smoked per day and depression and anxiety, there was no clear evidence for associations between the *CHRNA5-A3-B4* variant and these outcomes in smokers. If heavier smoking were to cause depression or anxiety, we would expect to see an increased risk of depression or anxiety per copy of the minor allele of rs16969968/rs1051730, which increases smoking heaviness, in current smokers and potentially also in former smokers, but no difference in risk for never smokers. In our meta-analyses, ORs for the effect of rs16969968/rs1051730 in current, former and ever smokers were all close to the null, with the CIs for these estimates all overlapping the null. In addition, we found no evidence to suggest that the variant was differentially associated with depression or anxiety according to smoking status.

Our results are consistent with those of the two previous Mendelian randomisation studies, which did not find evidence that smoking increases depression or anxiety[40] or ante-natal depression.[41] Both of these studies were included in this meta-analysis and the HUNT study made up more than half of the study sample in some analyses. However, exclusion of either of these samples did not make a substantial difference to effect estimates (see online supplementary figures S6 and S7). These findings suggest that previous findings linking smoking to higher levels of depression and anxiety[2 9 19 44] may be due to residual confounding, a shared vulnerability to both mental disorders and

smoking behaviour,[45] or reverse causality (eg, if smokers smoke in an attempt to alleviate the symptoms of depression or anxiety or depressed smokers find it more difficult to quit). Numerous longitudinal studies have reported that depressive symptoms in childhood and adolescence are associated with increased risk of smoking initiation or progression to tobacco dependence.[2 17 19 46–48] At the same time, smoking cessation appears to be associated with an acute increase in depressive symptoms among a subgroup of smokers, and these individuals have poor smoking cessation outcomes.[18] Taken together, this suggests that people experiencing depressive symptoms may smoke (or relapse to smoking) in an attempt to self-medicate these symptoms. A Mendelian randomisation study of the association of genetic variants for depression and anxiety with smoking behaviour would be required to investigate the self-medication hypothesis. However, genetic variants robustly associated with depression and anxiety have yet to be identified.[49]

Some caution should be taken in completely ruling out an effect of this variant on depression and anxiety within this analysis. The CIs for the associations of rs16969968/rs1051730 with depression and anxiety in current smokers overlap the estimates for the per cigarette per day increase in ORs of depression and anxiety, so we cannot conclusively say that the Mendelian randomisation analysis results differ from the observational results (figure 3). This is the most direct comparison that we can make with observational estimates in our data, given that the minor allele of rs16969968/rs1051730 is associated with an average of one cigarette per day increase in smoking heaviness.[34] However, this comparison may be problematic because cigarettes per day, a self-reported measure of tobacco exposure, does not take into account variation in smoking topography, such as the amount of a cigarette an individual smokes or the depth of inhalation.[50] The *CHRNA5-A3-B4* variant is an instrument for lifetime tobacco exposure within current smokers, and this is not fully captured by cigarettes per day. It has been shown that rs16969968/rs1051730 explains more of the variance in an objective measure of tobacco exposure, cotinine (4%), than in self-reported cigarettes per day (1%).[31 35] This appears to be why the variant shows a much stronger association with lung cancer than predicted from the observed associations with self-reported cigarettes per day.[31] Therefore if higher levels of smoking did cause depression or anxiety, we might expect the effects of rs16969968/rs1051730 to be considerably larger than those seen observationally per cigarette per day. For the same reason, we did not perform instrumental variable analysis to estimate the magnitude of the causal effect of smoking heaviness on depression or anxiety. It has been demonstrated that using cigarettes per day as an intermediate variable in Mendelian randomisation analyses using rs16969968/rs1051730 can lead to large biases in causal effect size estimates.[51]

The rs16969968/rs1051730 variant associates with smoking heaviness within smokers but is not an instrument for smoking status (ever smoking vs never smoking).[29] Therefore we cannot rule out the possibility that being a smoker, rather than smoking heaviness could influence likelihood of depression or anxiety. However, we do see an observational association between smoking heaviness and depression and anxiety (figure 3), and a dose-dependent relationship between an exposure and an outcome strengthens support for a causal association.[52] We might therefore expect to observe an association between rs16969968/rs1051730 and depression or anxiety if smoking were to cause these conditions. Furthermore, while rs16969968/rs1051730 is the strongest genetic contributor to smoking behaviour identified to date,[53] this variant only explains a fraction of the estimated 50% of total variation in smoking behaviour within a population at any one time that is due to genetic factors.[54] Further signals for smoking heaviness have been identified in the same gene cluster,[55] in other nicotinic receptor units and in other genes, such as those related to nicotine metabolism like *CYP2A6*.[56] Combining these variants in genetic risk scores for smoking behaviour in Mendelian randomisation studies will be an important future direction for validation of these results.

We would not expect to see an effect of rs16969968/rs1051730 on depression or anxiety in never smokers, because the variant is not associated with smoking heaviness within these individuals. Thus, this group can be used to test potential bias due to pleiotropy (that the gene affects more than one exposure) in Mendelian randomisation analyses.[38] We did observe some evidence for an association between the variant and anxiety in never smokers, a finding previously reported by the HUNT study.[40] Removal of HUNT from this analysis did not affect the point estimate, suggesting that this association is not driven solely by the data from this study (see online supplementary figure S6). However, using case definition 2 (where available) for anxiety in preference to case definition 1 slightly attenuated this association in never smokers (see online supplementary figure S8). While this may be a chance finding, it is possible that rs16969968/rs1051730 or a variant in linkage disequilibrium with this variant, may affect anxiety directly, not through tobacco consumption. There is some suggestion from animal studies that nicotinic acetylcholine receptors may play a role in anxiety (eg, mice lacking the α4 subunit show increases in anxiety-related behaviour[57]). However, there is currently little evidence for this association in humans, and rs16969968/rs1051730 has not been identified in genome-wide association studies of depression or anxiety to date.[49 58 59]

It is important to note that stratifying by the measured exposure variable in Mendelian randomisation studies can lead to collider bias.[60 61] In this specific analysis, if both the genetic variant and anxiety cause individuals to smoke, then stratifying on smoking could, in theory,

induce an association between the variant and anxiety.[13] We do not think that collider bias is likely to be a major issue in these analyses because rs16969968/rs1051730 does not appear to be associated with smoking initiation in this sample (see online supplementary figure S9) or in previous studies.[29 30] There is, however, as reported previously in a few specific populations[36 62 63] some evidence that the minor allele of rs16969968/rs1051730 is associated with smoking cessation; ORs of being a current compared to a former smoker were 1.08-fold higher per copy of the minor allele (95% CI 1.06 to 1.11) in this sample.

## Strengths and limitations

The key strength of this study is the large sample size, using data on over 125 000 individuals from 25 different populations. Despite this, we did not have the power to rule out the possibility of a causal effect. A substantial increase in sample size would be required to be confident that what we observe is a true null association in smokers. We hope that our estimates may be combined with those of further studies addressing the same question in future meta-analyses, to provide more definitive answers.

One of the main limitations is the use of broad definitions of depression and anxiety rather than clinical definitions, which were not available in all studies. It is possible that use of more precise phenotypic measures of depression and anxiety based solely on clinical criteria could yield stronger results because non-differential misclassification of a binary outcome is likely to attenuate associations towards the null.[2 64] However, we showed the expected observational associations between smoking and depression, anxiety and psychological distress. In addition, we used two case definitions and performed a sensitivity analysis using case definition 2 in preference (where both were available) which produced similar results (see online supplementary figure S8). Sensitivity analyses performed excluding lifetime definitions of depression or anxiety also produced similar results (see online supplementary figure S10). Finally, restricting our analyses to those studies with questionnaires based on clinical criteria or self-report of doctor diagnosis produced consistent results, although these analyses were underpowered (see online supplementary figure S11).

Although we analysed depression and anxiety separately, these conditions are highly comorbid in the general population[65 66] and symptom scale questionnaires are not adequate to distinguish between them.[67 68] In addition, the definition of anxiety we used encompassed general anxiety disorder, panic and phobias. It is possible that these conditions have different aetiologies.[9] Therefore, we cannot make inferences about specific anxiety disorders from these results. Furthermore, the sample encompasses a wide age range, so it is unlikely that this analysis would be able to capture any age-specific effects of smoking on depression and anxiety.

## CONCLUSION

In conclusion, our Mendelian randomisation analyses do not support a causal role of smoking heaviness among smokers in the development of depression and anxiety. While we cannot directly address the question of whether smoking initiation (ie, starting smoking) plays a causal role in relation to these outcomes, we expect that if it did we would also see a dose-dependent relationship between smoking heaviness and depression and anxiety. We see such an association in our observational analyses, but no strong evidence for this in our Mendelian randomisation analyses. Future research should focus on the possible role of depression and anxiety in increasing susceptibility to smoking. As larger genome-wide association studies of depression and anxiety emerge, it is likely that genetic variants will be identified that can be utilised in Mendelian randomisation analyses for this purpose.

**Author affiliations**
[1]MRC Integrative Epidemiology Unit (IEU) at the University of Bristol, UK
[2]UK Centre for Tobacco and Alcohol Studies, School of Experimental Psychology, University of Bristol, Bristol, UK
[3]Department of Public Health, Faculty of Medicine, Norwegian University of Science and Technology, Trondheim, Norway
[4]Forensic Department, Research Centre Bröset St. Olav's University Hospital Trondheim, Trondheim, Norway
[5]Faculty of Medicine, Department of Laboratory Medicine, Children's and Women's Health, Norwegian University of Science and Technology, Trondheim, Norway
[6]Centre for Cognitive Ageing and Cognitive Epidemiology, University of Edinburgh, Edinburgh, UK
[7]Medical Genetics Section, Centre for Genomic and Experimental Medicine, Institute of Genetics and Molecular Medicine, University of Edinburgh, Edinburgh, UK
[8]Queensland Brain Institute, The University of Queensland, Brisbane, QLD, Australia
[9]Department of Epidemiology and Public Health, University College London, London, UK
[10]Department of Epidemiology, Erasmus Medical Center, Rotterdam, The Netherlands
[11]University of Helsinki, Hjelt institute, Helsinki, Finland
[12]University of Eastern Finland, Institute of Public Health & Clinical Nutrition, Kuopio, Finland
[13]Department of Chronic Disease Prevention, National Institute for Health and Welfare, Helsinki, Finland
[14]Hospital District of North Karelia, Joensuu, Finland
[15]Department of Mental Health and Substance Abuse Services, National Institute for Health and Welfare, Helsinki, Finland
[16]Institute of Health Sciences, FI-90014 University of Oulu, Finland
[17]Biocenter Oulu, FI-90014 University of Oulu, Finland
[18]South West London and St George's Mental Health Trust, London, UK
[19]Population, Policy and Practice, UCL Institute of Child Health, University College London, UK
[20]Research Centre for Prevention and Health, the Capital Region of Denmark, Denmark
[21]Metabolic Genetics Section, Faculty of Health and Medical Sciences, Novo Nordisk Foundation Centre for Basic Metabolic Research, University of Copenhagen, Copenhagen, Denmark
[22]Copenhagen Prospective Studies on Asthma in Childhood, Faculty of Health and Medical Sciences, University of Copenhagen, Copenhagen, Denmark.
[23]Danish Pediatric Asthma Center, Gentofte Hospital, The Capital Region, Copenhagen, Denmark.
[24]Institute of Public Health and Center for Healthy Aging, University of Copenhagen, Copenhagen, Denmark

[25]Department of Biological Psychology, Netherlands Twin Register, VU University, Amsterdam, The Netherlands
[26]Department of Internal Medicine, Lausanne University Hospital, Lausanne, Switzerland
[27]Faculty of Epidemiology and Population Health, London School of Hygiene & Tropical Medicine, London, UK
[28]Department of Primary Care & Population Health, UCL, London, UK
[29]Institute of Behavioural Sciences, University of Helsinki, Helsinki, Finland
[30]Folkhälsan Research Centre, Helsinki, Finland
[31]Wellcome Trust Sanger Institute, Cambridge, UK
[32]The Medical and Population Genomics Program, The Broad Institute of MIT and Harvard, Cambridge, Massachusetts, USA
[33]Institute for Molecular Medicine Finland (FIMM), University of Helsinki, Finland
[34]MRC Unit for Lifelong Health, Ageing at UCL, UK
[35]School of Social and Community Medicine, University of Bristol, Bristol, UK
[36]Department of Psychology, University of Toronto, Toronto, Canada
[37]Rotman Research Institute, Toronto, Canada
[38]Department of Psychological Medicine, University of Otago, Christchurch, New Zealand
[39]Childhood Cancer Research Group, University of Oxford, Oxford, UK
[40]Department of Oncology, University of Oxford, Oxford, UK
[41]Department of Pathology, University of Otago, Christchurch, New Zealand
[42]Departments of Physiology and Nutritional Sciences, University of Toronto, Toronto, Canada
[43]Hospital for Sick Children, Toronto, Canada
[44]Departments of Psychology and Psychiatry, University of Toronto, Toronto, Canada
[45]Institute for Clinical Research, University of Southern Denmark, Odense, Denmark
[46]Department of Medical Genetics, University of Helsinki and University Central Hospital, Helsinki, Finland
[47]National Institute for Health and Welfare, Finland
[48]Department of General Practice and Primary health Care, University of Helsinki, Finland
[49]Unit of General Practice, Helsinki University Central Hospital, Helsinki, Finland
[50]Vasa Central Hospital, Vasa, Finland
[51]Institute of Cardiovascular Science, University College London, UK
[52]Department of Psychiatry, Lausanne University Hospital, Prilly, Switzerland
[53]Department of Clinical Experimental Research, Glostrup University Hospital, Denmark
[54]Faculty of Health and Medical Sciences, Department of Clinical Medicine, University of Copenhagen, Denmark
[55]School of Population Health and Sansom Institute, University of South Australia, Adelaide, Australia
[56]South Australian Health and Medical Research Institute, Adelaide, Australia
[57]Department of Psychiatry, Oulu University Hospital, Oulu, Finland
[58]Department of Epidemiology and Biostatistics, MRC Health Protection Agency (HPA) Centre for Environment and Health, School of Public Health, Imperial College London, UK
[59]Unit of Primary Care, Oulu University Hospital, Oulu, Finland
[60]Department of Children and Young People and Families, National Institute for Health and Welfare, Oulu, Finland
[61]Department of Epidemiology and Psychiatry, Erasmus Medical Center, Rotterdam, The Netherlands
[62]Medical Research Council Human Genetics Unit, Institute of Genetics and Molecular Medicine, University of Edinburgh, Edinburgh, UK

**Acknowledgements** ALSPAC: The authors are extremely grateful to all the families who took part in this study, the midwives for their help in recruiting them, and the whole ALSPAC team, which includes interviewers, computer and laboratory technicians, clerical workers, research scientists, volunteers, managers, receptionists and nurses. BRHS: The British Regional Heart Study is a British Heart Foundation (BHF) Research Group. BWHHS: The authors thank all BWHHS participants, the general practitioners and their staff who have supported data collection since the study inception. CaPS: The Caerphilly Prospective Study was conducted by the former MRC Epidemiology Unit (South Wales). The Caerphilly archive is now maintained by the School of Social and Community Medicine in Bristol University. The authors thank the Health and Social Care Information Centre (HCSIC) for helping us maintain long term follow-up with the cohort. They also thank all the men who have given their time to be participants in CaPS. HUNT: Nord-Trøndelag Health Study (The HUNT Study) is a collaboration between HUNT Research Centre (Faculty of Medicine, Norwegian University of Science and Technology NTNU), Nord-Trøndelag County Council and the Norwegian Institute of Public Health. NFBC: The authors thank the late Professor Paula Rantakallio (launch of NFBCs), and Ms Outi Tornwall and Ms Minttu Jussila (DNA biobanking). The authors would like to acknowledge the contribution of the late Academian of Science Leena Peltonen. NSHD: The authors are very grateful to the members of this birth cohort for their continuing interest and participation in the study.

**Collaborators** ALSPAC (AET, MEF, MRM, GDS), HUNT ( JHB, PRR, MEG, FS), Generation Scotland (REM, AC, DJP, CH), Whitehall/ELSA (MeK, JE, MiK), Rotterdam (HT, SSM), FINRISK (AL, TL, TP, TK), NFBC (M-RJ, MaK, JV, FD), 1958BC (AC, CP, EH), Health2006/Health2008/Inter99 (LLNH, RKJ, TS, JFE, TSA, AL, ELM), NTR (CCM, GW, DIB, JMV), COLAUS/PsyCoLaus (PM-V, MP), BWHHS (CED, AA, JPC), BRHS (RWM, LTL), HBCS ( JL, AP, JGE, KR), NSHD (AW, DK), Goya females (EAN, LP), CaPS (YB-S), SYS-P (AP-YW, ZP, TP), CHDS (LJH, MAK), Patch 2 (ECJ, MM).

**Contributors** MRM and GDS conceived the study. AET and MEF drafted the analysis protocol and co-ordinated the data analysis. AET conducted the final meta-analyses. AET and MRM drafted the initial manuscript. JHB, MEG, FS, REM, AC, JE, SSM, AL, TL, TP, MK, FD, AC, LLNH, TSA, RKJ, TS, JFE, ELM, CCM, JMV, GW, PMV, CED, AA, LTL, JL, AP, KR, AW, LP, APYW and LJH all performed analyses within individual studies. MM, ECJ, MAK, ZP, TP, YBS, EAN, DK, MiK, JGE, RWM, JPC, MP, DIB, AL, CP, EH, JV, MRJ, TK, HT, MeK, DJP, CH and PRR all substantially contributed to data acquisition in contributing studies. All authors commented on a draft of the manuscript and approved the final version.

**Funding** 1958BC: Statistical analyses were funded by the Academy of Finland (Project 24300796 and SALVE/PREVMEDSYN). DNA collection was funded by MRC grant G0000934 and cell-line creation by Wellcome Trust grant 068545/ Z/02. This research used resources provided by the Type 1 Diabetes Genetics Consortium, a collaborative clinical study sponsored by the National Institute of Diabetes and Digestive and Kidney Diseases (NIDDK), National Institute of Allergy and Infectious Diseases, National Human Genome Research Institute, National Institute of Child Health and Human Development, and Juvenile Diabetes Research Foundation International ( JDRF) and supported by U01 DK062418. Funding for the project was provided by the Wellcome Trust under the award 076113. Great Ormond Street Hospital/University College London, Institute of Child Health receives a proportion of funding from the Department of Health's National Institute for Health Research (NIHR) ('Biomedical Research Centres' funding). ALSPAC: The UK Medical Research Council and the Wellcome Trust (Grant ref: 092731) and the University of Bristol provide core support for ALSPAC. This work was supported by the Wellcome Trust (grant number 086684) and the Medical Research Council (grant numbers MR/J01351X/1, G0800612, G0802736, MC_UU_12013/1, MC_UU_12013/6). BRHS: The collection and management of data over the last 34 years of the BRHS has been made possible through grant funding from UK government agencies and charities. BWHHS: The British Women's Heart and Health Study has been supported by funding from the British Heart Foundation (BHF) (grant PG/09/022) and the UK Department of Health Policy Research Programme (England) (grant 0090049). The BWHHS HumanCVD data were funded by the BHF (PG/07/131/24254). CHDS: The Christchurch Health and Development Study has been supported by funding from the Health Research Council of New Zealand, the National Child Health Research Foundation (Cure Kids), the Canterbury Medical Research Foundation, the New Zealand Lottery Grants Board, the University of Otago, the Carney Centre for Pharmacogenomics, the James Hume Bequest Fund, US NIH grant MH077874, and NIDA grant "A developmental model of gene-environment interplay in SUDs" (R01DA024413) 2007–2012. Colaus/PsyCoLaus: The CoLaus/PsyCoLaus study was supported by four grants of the Swiss National Science Foundation (#105993, 118308, 139468 and 122661), two unrestricted grants from GlaxoSmithKline as well as by the Faculty of Biology and Medicine of the Lausanne University. ELSA: ELSA is funded by the National Institute on Aging in the US (R01 AG017644;R01AG1764406S1) and by a consortium of UK Government departments (including: Department for

Communities and Local Government, Department for Transport, Department for Work and Pensions, Department of Health, HM Revenue and Customs and Office for National Statistics). FINRISK: This study was supported by the Academy of Finland Center of Excellence in Complex Disease Genetics (grant numbers 213506, 129680), the Academy of Finland (grant numbers 139635, 129494, 136895, 263836 and 141054), the Sigrid Juselius Foundation, and ENGAGE – European Network for Genetic and Genomic Epidemiology, FP7-HEALTH-F4-2007, grant agreement number 201413. Generation Scotland: Generation Scotland has received core funding from the Chief Scientist Office of the Scottish Government Health Directorates CZD/16/6 and the Scottish Funding Council HR03006. Genotyping of the GS:SFHS samples was carried out by the Genetics Core Laboratory at the Wellcome Trust Clinical Research Facility, Edinburgh, Scotland and was funded by the UK's Medical Research Council. GOYA females: The GOYA study was conducted as part of the activities of the Danish Obesity Research Centre (DanORC, www.danorc.dk) and The MRC centre for Causal Analyses in Translational Epidemiology (MRC CAiTE). The genotyping for GOYA was funded by the Wellcome Trust (WT 084762). GOYA is a nested study within The Danish National Birth Cohort which was established with major funding from the Danish National Research Foundation. Additional support for this cohort has been obtained from the Pharmacy Foundation, the Egmont Foundation, The March of Dimes Birth Defects Foundation, the Augustinus Foundation, and the Health Foundation. HBCS: Helsinki Birth Cohort Study has been supported by grants from the Academy of Finland, the Finnish Diabetes Research Society, Folkhälsan Research Foundation, Novo Nordisk Foundation, Finska Läkaresällskapet, Signe and Ane Gyllenberg Foundation, University of Helsinki, Ministry of Education, Ahokas Foundation, Emil Aaltonen Foundation. NFBC: NFBC1966 and NFBC1986 received financial support from the Academy of Finland (project grants 104781, 120315, 129269, 1114194, 24300796, 141042 Center of Excellence in Complex Disease Genetics and SALVE), University Hospital Oulu, Biocenter, University of Oulu, Finland (75617), NHLBI grant 5R01HL087679-02 through the STAMPEED program (1RL1MH083268-01), NIH/NIMH (5R01MH63706:02), the European Commission (EURO-BLCS, Framework 5 award QLG1-CT-2000-01643), ENGAGE project and grant agreement HEALTH-F4-2007-201413, EU FP7 EurHEALTHAgeing -277849, the Medical Research Council, UK (G0500539, G0600705, G1002319, PrevMetSyn/SALVE) and the MRC, Centenary Early Career Award. The DNA extractions, sample quality controls, biobank up-keeping and aliquotting was performed in the National Public Health Institute, Biomedicum Helsinki, Finland and supported financially by the Academy of Finland and Biocentrum Helsinki. NSHD: This work was funded by the Medical Research Council [MC_UU_12019/1]. NTR: This study was supported by the European Research Council (ERC Starting Grant 284167 PI Vink), Netherlands Organization for Scientific Research (NWO: MagW/ZonMW grants 904-61-090, 985-10-002, 904-61-193, 480-04-004, 400-05-717, Addiction-31160008 Middelgroot-911-09-032, Spinozapremie 56-464-14192), BBRMI-NL (Biobanking and Biomolecular Resources Research Infrastructure), VU University's Institutes for Health and Care Research and Neuroscience Campus Amsterdam. Patch 2: The Patch 2 study was funded by the Imperial Cancer Research Fund (ICAF) and Cancer Research UK (CRUK) Programme Grants to the General Practice Research Group at the University of Oxford. Rotterdam: The Rotterdam Study was supported by the Erasmus Medical Center and the Erasmus University Rotterdam, the Netherlands Organisation of Scientific Research (NWO), the Netherlands Organisation for Health Research and Development (ZonMw), the Ministry of Education, Culture, and Science, the Ministry of Health, Welfare, and Sports, the European Commission (DG-XII). SYS-P: The SYS-P is funded by the Canadian Institutes of Health Research, the Canadian Foundation for Innovation, and the Heart and Stroke Foundation of Ontario. Whitehall: The Whitehall II study has been supported by grants from the Medical Research Council; British Heart Foundation; Health and Safety Executive; Department of Health; National Heart Lung and Blood Institute (NHLBI: HL36310) and National Institute on Aging (AG13196), US, NIH; Agency for Health Care Policy Research (HS06516); and the John D and Catherine T MacArthur Foundation Research Networks on Successful Midlife Development and Socio-economic Status and Health.

**Competing interests** MRM: Medical Research Council, LTL: British Heart Foundation, LJH: New Zealand Health Research Council, MK: Jim and Mary Carney Charitable Trust, New Zealand Health Research Council, CCM: European Research Council, DIB, JMV: NWO, European Research Council,

National Institutes of Health, HT: Dutch Medical Research Council, TP: Canadian Institutes of Health Research, MK: Medical Research Council, NIH Heart, Lung and Blood Institute and Economic and Social Research Council); financial relationships with any organisations that might have an interest in the submitted work in the previous 3 years (LJH: Australian National Health and Medical Research council, PMV: Swiss National Science Foundation, MP: Eli Lilly and Lundbeck advisory board membership, Swiss National Science Foundation, TK: Pfizer consultancy on tobacco dependence, Juho Vainio Foundation, The Finnish Medical Society, TSA: Gene Diet Interactions in Obesity project, HT: KOC University, Pfizer), YBS: grants from NIHR, MRC, PDUK, BHF. AET, MRM and MEF are members of the UK Centre for Tobacco and Alcohol Studies, a UKCRC Public Health Research: Centre of Excellence. Funding from British Heart Foundation, Cancer Research UK, Economic and Social Research Council, Medical Research Council, and the National Institute for Health Research, under the auspices of the UK Clinical Research Collaboration is gratefully acknowledged. LTL is supported by BHF programme grant RG/08/013/25942. LP is funded by an MRC Population Health Scientist Fellowship (MR/J012165/1). TSA was supported by the Gene Diet Interactions in Obesity (GENDINOB, www.gendinob.dk) postdoctoral fellowship grant. LLNH was supported by the Health Insurance Foundation (grant No. 2010 B 131). The work of SSM was funded by the Netherlands Genomics Initiative (NGI)/Netherlands Organization for Scientific Research (NWO) (grant-numbers 050-060-810 NCHA); the work of HT was supported by NWO-VIDI (grant 017-106-370). MeK is partially supported by the Economic and Social Research Council International Centre for Life Course Studies in Society and Health (RES-596-28-0001).

**Patient consent** Obtained.

**Ethics approval** 1958BC: Ethics approval for the study was obtained from the South-East Multi-Centre Research Ethics Committee (Ref: 01/1/44) and the Joint UCL/UCLH Committees on the Ethics of Human Research (Committee A) Ref: 08/H0714/40. ALSPAC: Ethics approval for the study was obtained from the ALSPAC Ethics and Law Committee and the Local Research Ethics Committee. BRHS: The BRHS has local (from each of the districts in which the study was based) and multicentre ethical committee approvals. Ethical approval was granted for the BWHHS from the London Multi-Centre Research Ethics Committee and 23 Local Research Ethics Committees. BWHHS: Ethical approval was granted for the BWHHS from the London Multi-Centre Research Ethics Committee and 23 Local Research Ethics Committees. CaPS: Ethics approval was obtained from the South Glamorgan Area Health Authority, the Gwent REC, and the South Wales Research Ethics Committee D. CHDS: All phases of the study have received ethical approval from the regional Health and Disability Ethics Committee. Colaus/PsyCoLaus: Colaus and PsyCoLaus were approved by the Institutional Ethic's Committee of the University of Lausanne. ELSA: ELSA has been approved by the National Research Ethics Service. FINRISK: The 2002 and 2007 FINRISK surveys have been approved by the Coordinating Ethics Committee of the Helsinki University Hospital District. Generation Scotland: Ethical approval for the study was given by the NHS Tayside committee on research ethics (reference 05/s1401/89). GOYA females: The study was approved by the regional scientific ethics committee and by the Danish Data Protection Board. HBCS: The research plan of the HBCS was approved by the Institutional Review Board of the National Public Health Institute. Health2006/Health2008/Inter99: The studies have been approved by the Ethical Committee of Copenhagen. HUNT: Use of data in the present study was approved by the Regional Committee for Medical Research Ethics (Reference nr. 2013/1127/REK midt). NFBC: The University of Oulu Ethics Committee and the Ethical Committee of Northern Ostrobothnia Hospital District have approved the study. NHANES: Data collection for NHANES was approved by the NCHS Research Ethics Review Board. Analysis of de-identified data from the survey is exempt from the federal regulations for the protection of human research participants. Analysis of restricted data through the NCHS Research Data Center is also approved by the NCHS ERB. NSHD: Ethical approval was given by the Central Manchester Research Ethics Committee. NTR: The NTR study was approved by the Central Ethics Committee on Research Involving Human Subjects of the VU University Medical Center, Amsterdam (IRB number IRB-2991 under Federalwide Assurance 3703; IRB/institute code 03-180). Patch 2: Ethical approval was obtained from the Anglia and Oxford Multicentre Research Ethics Committee and from the Local Research Ethics Committees covering the areas of

residence of the patients. Rotterdam: The Medical Ethics Committee of Erasmus Medical Center Rotterdam approved the study. SYS-P: The Research Ethics Committee of the Chicoutimi Hospital in Quebec, Canada approved the study protocol. Whitehall: Ethical approval for the Whitehall II study was obtained from the University College London Medical School committee on the ethics of human research.

**Provenance and peer review** Not commissioned; externally peer reviewed.

**Data sharing statement** 1958BC: This study makes use of data generated by the Wellcome Trust Case-Control Consortium. A full list of investigators who contributed to generation of the data is available from the Wellcome Trust Case-Control Consortium website. The 1958 birth cohort data can be accessed via the UK Data Service (http://ukdataservice.ac.uk/). ALSPAC: Data used for this submission was made available on request to the ALSPAC executive committee (alspac-exec@bristol.ac.uk). The ALSPAC data management plan (available here:http://www.bristol.ac.uk/alspac/researchers/data-access/) describes in detail the policy regarding data sharing, which is through a system of managed open access. BRHS: We welcome proposals for collaborative projects and data sharing (http://www.ucl.ac.uk/pcph/research-groups-themes/brhs-pub). For general data sharing enquiries, please contact Lucy Lennon (l.lennon@ucl.ac.uk). BWHHS: All BWHHS data collected is held by the research team based at London School of Hygiene and Tropical Medicine, for ongoing analysis. If you would like to collaborate with the BWHHS team, contact the study coordinator, AA (antoinette.amuzu@lshtm.ac.uk). Data and biological samples provided to the collaborators can only be used for the purposes originally stated and must not be used in any other way without re-application to the BWHHS team. No data should be passed on to any third party unless they were specified in the original application. CaPS: Data used for the Caerphilly Prospective study (CaPS) was made available by the CaPS access committee (Chair: Professor Kay Tee Khaw). More infomation about its managed access procedure is available on the study website (http://www.bris.ac.uk/social-community-medicine/people/project/1392). CHDS: Data contributed for this submission are available on request from the CHDS (john.horwood@otago.ac.nz). Colaus/PsyCoLaus: Data from the CoLaus/PsyCoLaus study can be requested according to the procedure described on the CoLaus website (http://www.colaus.ch/en/cls_home/cls_pro_home/cls-research-3.htm). ELSA: ELSA data are made available through the ESDS website (http://www.elsa-project.ac.uk/availableData). FINRISK: Data used for this submission will be made available on request to the FINRISK Management Group, according to the given ethical guidelines and Finnish legislation. Generation Scotland: Data is available on request (access@generationscotland.org). GOYA females: An anonymized copy of the data used for this submission will be made available on request to the GOYA analysts after permission have been given by the DNBC executive committee (www.dnbc.dk). HBCS: Data used for this submission will be made available on request to the HBCS executive committee (johan.eriksson@helsinki.fi). Health2006/Health2008/Inter99: Data used for this submission can be made available on request to the Research Centre for Prevention and Health (http://www.regionh.dk/fcfs/Menu/). Please contact LLNH (lise.lotte.nystrup.husemoen@regionh.dk) or AL (allan.linneberg@regionh.dk). HUNT: Data used from the HUNT Study for this submission will be made available on request to the HUNT Data Access Committee (hunt@medisin.ntnu.no). The HUNT data access information (http://www.ntnu.edu/hunt/data) describes in detail the policy regarding data availability. NFBC: Data used for this submission can be made available on request to Tuula Ylitalo (tuula.ylitalo@oulu.fi), Minna Mannikko (minna.annikko@oulu.fi) or M-RJ (m.jarvelin@imperial.ac.uk). NHANES: NHANES data can be accessed here: (http://www.cdc.gov/nchs/nhanes.htm). The genotype used in this analysis is a restricted variable. Applications for access to these data must be made through the Research Data Center: (http://www.cdc.gov/rdc/). NSHD: The NSHD data are made available to researchers who submit data requests (tomrclha.swiftinfo@ucl.ac.uk). More information is available in the full policy documents (http://www.nshd.mrc.ac.uk/data.aspx). Managed access is in place for this study to ensure that use of the data are within the bounds of consent given previously by participants, and to safeguard any potential threat to anonymity since the participants are all born in the same week. NTR: Data used for this submission will be made available on request to the NTR committee (ntr@psy.vu.nl). SYS-P: Data used for this submission will be made available on request to the SYS principal investigators (http://www.

saguenay-youth-study.org) (scientific community/collaborative interests). Whitehall: Data from the Whitehall II study are made publicly available as described in the Whitehall II data sharing policy (http://www.ucl.ac.uk/whitehallII/datasharing).

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
