## [Reviewer comments · BMJ Open]

This paper was submitted to the BMJ but declined for publication following peer review. The authors addressed the reviewers' comments and submitted the revised paper to BMJ Open. The paper was subsequently accepted for publication at BMJ Open.

ARTICLE DETAILS

TITLE (PROVISIONAL)	Investigating the possible causal association of smoking with depression and anxiety using Mendelian randomisation meta-analysis: The CARTA consortium
AUTHORS	Amy E Taylor, Meg E Fluharty , Johan H Bjørngaard, Maiken Elvestad Gabrielsen, Frank Skorpen, Riccardo E Marioni, Archie Campbell, Jorgen Engmann, Saira Saeed Mirza, , Anu Loukola, Tiina Laatikainen, Timo Partonen , Marika Kaakinen, Francesca Ducci, Alana Cavadino, Lise Lotte N Husemoen, Tarunveer Singh Ahluwalia, Rikke Kart Jacobsen, Tea Skaaby, Jeanette Frost Ebstrup, Erik Lykke Mortensen, Camelia C Minica, Jacqueline M Vink, Gonneke Willemsen, Pedro Marques-Vidal, Caroline E Dale, Antoinette Amuzu, Lucy T Lennon, Jari Lahti, Aarno Palotie, Katri Räikkönen, Andrew Wong, Lavinia Paternoster, Angelita Pui-Yee Wong, , L John Horwood, Michael Murphy, Generation Scotland, Elaine C Johnstone, Martin A Kennedy, Zdenka Pausova ,Tomáš Paus, Yoav Ben-Shlomo, Ellen A Nohr, Diana Kuh, Mika Kivimaki, Johan G Eriksson, Richard W Morris, Juan P Casas, Martin Preisig, Dorret I Boomsma, Allan Linneberg, Chris Power, Elina Hyppönen , Juha Veijola, Marjo- Riitta Jarvelin, Tellervo Korhonen, Henning Tiemeier, Meena Kumari, David J Porteous, Caroline Hayward, Pål R Romundstad, George Davey Smith, Marcus R Munafò

VERSION 1 - REVIEW

REVIEWER	Burgess, Stephen University of Cambridge, Department of Public Health and Primary Care
REVIEW RETURNED	27-May-2014

GENERAL COMMENTS	This is an interesting study with the potential to answer an important question about the direction of causation between smoking and psychological disorders. There are two main limitations: 1) Power - Null conclusions from Mendelian randomization studies are typically limited by power. This is because genetic variants do not explain much of the variation in risk factors. In this case, the expected per allele genetic association of the variant with the outcome based on the observational association was an odds ratio
---

of 1.03. There are reasons one may expect a stronger genetic association than the observational association if there were a true causal link; however, these are speculative and depend on the function of the genetic variant. The genetic associations observed are therefore compatible with the (positive) observational associations, meaning that the null result could reflect a lack of power to detect a true effect. On the other hand, a more conclusive Mendelian randomization study addressing this question is unlikely to be performed in the medium- to long-term future. This analysis included a genetic variant in the gene region having by far the strongest association with smoking-related exposures, and have a total sample size of 125,000+. The addition of even 20,000 participants (say) is highly unlikely to change the results of this investigation substantially.

2) Function of the genetic variant - The analysis uses a single genetic variant. While this has definite benefits in terms of plausibility of the Mendelian randomization assumptions (the specific association of the variant with smoking-related exposures has scientific merit, and assessing the instrumental variable assumptions is only required for a single variant), there are also points to take note. In particular, the genetic variant assesses the causal nature of the aspects of smoking-related exposures associated with the variant. For example, the strongest effect of the variant is on cotinine levels. Is it plausible that cotinine is the major mediator of the genetic association with depression/anxiety? Or could we have missed the true cause linking smoking with depression/anxiety?

3) Another limitation in terms of impact, is that these data (and hence the results) overlap largely with reference 38.

Major points:

1. The statement "...in a British cohort, the rs16969968 variant was associated with decreased depression [39] during pregnancy in women who smoked prior to pregnancy." is odd, especially in conjunction with the further statement "These findings are not consistent with a causal role of smoking in increasing depression or anxiety, but are inconclusive with respect to smoking decreasing depression." At face value, the association of the rs16969968 variant with decreased depression is entirely relevant to the discussion at hand. The conclusion that this finding (which is similar to that assessed by the investigators in this paper) is "inconclusive with respect to smoking decreasing depression" therefore requires some more unpacking. While I have read the authors' explanation, there is an element to me of the authors picking a plausible story to fit the data, rather than the conclusion coming clearly from the data. Either way, this statement is currently difficult to understand in the context of the paper, and requires some unwrapping.

2. "Therefore if higher levels of smoking did cause depression or anxiety, we might expect the effects of rs16969968/rs1051730 to be considerably larger than those seen observationally per cigarette per day." As stated above, this is speculative (but I wouldn't remove the sentence for this reason alone), and also it only makes sense if cotinine is the likely mediator of the genetic association of smoking with psychological disorders. Is this plausible?

I'd like to see a graph of associations of the genetic variant with smoking related variables - the per allele associations with cigarettes per day, smoking onset, smoking duration, cotinine levels etc. I think this would make clearer the true interpretation of the genetic association with the outcome, in terms of showing what the gene does and so what we are comparing in the comparison of the genetic groups. It would also be good to include other non-smokingrelated

	variables in such a graph, to assess the instrumental variable assumptions that the genetic variant is specifically associated with smoking-related exposures. Minor points: 3a. The forest plots will be unreadable at print size. Suggest a summarized figure comparing the "expected" genetic associations based on the observational evidence, with the measured genetic associations. The forest plots could then become Supplementary Figures. 3b. Similar summarized figures could be provided for the sexspecific analyses, and for the sensitivity analyses omitting the HUNT study and the study of reference 39, so that the results were provided for reference (rather than "results not shown" - if the time has been taken to produce the results, it would seem that little additional effort would be required to present the results).
--	--

REVIEWER	Au Yeung, SL The University of Hong Kong, School of Public Health
REVIEW RETURNED	05-Jun-2014

GENERAL COMMENTS	This study investigates the causal relationship of smoking heaviness with depression and anxiety using Mendelian randomization meta analysis. The main strengths of this paper include the use of Mendelian randomization approach (using genetic variants linked to smoking heaviness) which are less susceptible to confounding and reverse causation commonly found in observational studies; and the large sample size, which is crucial for a meaningful Mendelian randomization analysis. As such, this paper may provide more credible evidence on this topic. The following are some suggestions which may improve the paper. Major comments For a valid Mendelian randomization analysis, certain assumptions have to be fulfilled. The authors should describe in more details how these assumptions were assessed in the Methods, or if they were assessed previously, corresponding references should be provided. The absence of pleiotropy, one of these assumptions, is not depicted in Figure 1. Mendelian randomization analysis can be implemented in two ways, 1) Using genetic polymorphisms as proxies of exposure; or 2) Instrumental variable analysis using genetic polymorphisms as instrument. There has been some discussion on the utility of these approaches recently (Methodological challenges in Mendelian randomization published in Epidemiology, 2014). Can the authors explain why they choose the first approach over the second approach in this study? The observational analysis is highly prone to confounding as only a few factors were adjusted for in this analysis. Therefore, even the result of a meta analysis is likely to be biased by methodological issues described in the Introduction. Can the authors adjust for more potential confounders such as sociodemographic factors in each of the study included? Minor comments As the Mendelian randomization analysis only showed smoking heaviness (not smoking status) not associated with depression and anxiety among smokers, the conclusion in the abstract should be rephrased for better clarity. Inconsistencies in referencing style in the reference list, and typos in the author (Davey Smith G instead of Smith GD)
--

VERSION 1 – AUTHOR RESPONSE

Reviewer: 1:

Power - Null conclusions from Mendelian randomization studies are typically limited by power. This is because genetic variants do not explain much of the variation in risk factors. In this case, the expected per allele genetic association of the variant with the outcome based on the observational association was an odds ratio of 1.03. There are reasons one may expect a stronger genetic association than the observational association if there were a true causal link; however, these are speculative and depend on the function of the genetic variant. The genetic associations observed are therefore compatible with the (positive) observational associations, meaning that the null result could reflect a lack of power to detect a true effect. On the other hand, a more conclusive Mendelian randomization study addressing this question is unlikely to be performed in the medium- to long-term future. This analysis included a genetic variant in the gene region having by far the strongest association with smoking-related exposures, and have a total sample size of 125,000+. The addition of even 20,000 participants (say) is highly unlikely to change the results of this investigation substantially.

We agree with the reviewers' comments. This is a limitation of the study, and we now discuss this in some detail. One page 19, we say:

“Despite this, we did not have the power to rule out the possibility of a causal effect. A substantial increase in sample size would be required to be confident that what we observe is a true null association in smokers. We hope that our estimates may be combined with those of further studies addressing the same question in future meta-analyses, to provide more definitive answers.”

Function of the genetic variant - The analysis uses a single genetic variant. While this has definite benefits in terms of plausibility of the Mendelian randomization assumptions (the specific association of the variant with smoking-related exposures has scientific merit, and assessing the instrumental variable assumptions is only required for a single variant), there are also points to take note. In particular, the genetic variant assesses the causal nature of the aspects of smoking-related exposures associated with the variant. For example, the strongest effect of the variant is on cotinine levels. Is it plausible that cotinine is the major mediator of the genetic association with depression/anxiety? Or could we have missed the true cause linking smoking with depression/anxiety?

Yes- we believe that cotinine is likely to be the major mediator of this association, but only because it is a marker of tobacco exposure, not through an independent pathway. The genetic variant we use is in a gene encoding a nicotinic receptor subunit. Therefore, it is unlikely to affect nicotine metabolism directly. Any effect on cotinine (a metabolite of nicotine) conferred by the variant is therefore likely to be due to differences in tobacco intake rather than nicotine metabolism. We have expanded on this point on page 10.

“The rs16969968 variant is functional and leads to an amino acid change (D398N) in nicotinic receptor alpha5 subunit protein 33. The minor (risk) allele of this variant is associated with an average increase in smoking amount of one cigarette per day in smokers, and even more strongly associated with increases in cotinine (a metabolite of nicotine) levels 31 34 35.

However, given the known role of the variant in altering receptor function 33, it is likely that the greater variance explained for cotinine levels is due to this measure better capturing total tobacco exposure, and not because the variant directly affects nicotine metabolism 31.”

Another limitation in terms of impact, is that these data (and hence the results) overlap largely with reference 38.

We agree, although our sample size is substantially larger, and therefore the estimates we report correspondingly more precise. In addition, the results without the HUNT study are provided in supplementary material (Figure S6).

The statement "...in a British cohort, the rs16969968 variant was associated with decreased depression [39] during pregnancy in women who smoked prior to pregnancy." is odd, especially in conjunction with the further statement "These findings are not consistent with a causal role of smoking in increasing depression or anxiety, but are inconclusive with respect to smoking decreasing depression." At face value, the association of the rs16969968 variant with decreased depression is entirely relevant to the discussion at hand. The conclusion that this finding (which is similar to that assessed by the investigators in this paper) is "inconclusive with respect to smoking decreasing

depression" therefore requires some more unpacking. While I have read the authors' explanation, there is an element to me of the authors picking a plausible story to fit the data, rather than the conclusion coming clearly from the data. Either way, this statement is currently difficult to understand in the context of the paper, and requires some unwrapping.

We have amended this sentence to read:

"These findings are not consistent with a causal role of smoking in increasing depression or anxiety"

"Therefore if higher levels of smoking did cause depression or anxiety, we might expect the effects of rs16969968/rs1051730 to be considerably larger than those seen observationally per cigarette per day." As stated above, this is speculative (but I wouldn't remove the sentence for this reason alone), and also it only makes sense if cotinine is the likely mediator of the genetic association of smoking with psychological disorders. Is this plausible?

As explained above, we think cotinine is likely to be the mediator of this association because it is a marker of tobacco consumption. The bias in effect sizes that can arise from using selfreported

measures of tobacco exposure (e.g., cigarettes per day) has been illustrated in a recent paper by VanderWeele and colleagues. We have added this to the discussion:

"It has been demonstrated that using cigarettes per day as an intermediate variable in Mendelian randomisation analyses using rs16969968/rs1051730 can lead to large biases in causal effect size estimates 49."

I'd like to see a graph of associations of the genetic variant with smoking related variables - the per allele associations with cigarettes per day, smoking onset, smoking duration, cotinine levels etc. I think this would make clearer the true interpretation of the genetic association with the outcome, in terms of showing what the gene does and so what we are comparing in the comparison of the genetic groups. It would also be good to include other non-smoking-related variables in such a graph, to assess the instrumental variable assumptions that the genetic variant is specifically associated with smoking-related exposures.

We do not think that this is necessary given the robust published evidence for the association between rs16969968/rs1051730 and smoking heaviness as assessed by cigarettes per day and cotinine (see Ware et al. 2011, Munafò et al. 2012). This variant does not show robust evidence for associations with other smoking phenotypes (e.g., smoking initiation). We have discussed the fact that this genetic variant is an instrument for smoking heaviness and not initiation in the manuscript. Producing a table of potential confounders would be difficult as likely confounders are not assessed in all or in the same way across the CARTA studies. In addition, there is good evidence from previous publications, including the HUNT study, that the variant is not associated with measured confounders. We have mentioned this in the introduction.

The forest plots will be unreadable at print size. Suggest a summarized figure comparing the "expected" genetic associations based on the observational evidence, with the measured genetic associations. The forest plots could then become Supplementary Figures.

We have amended the Figures. Full forest plots have been moved to supplementary material.

Similar summarized figures could be provided for the sex-specific analyses, and for the sensitivity analyses omitting the HUNT study and the study of reference 39, so that the results were provided for reference (rather than "results not shown" - if the time has been taken to produce the results, it would seem that little additional effort would be required to present the results).

We have added most of the sensitivity analyses to the supplementary material. Sex-stratified analyses are not shown. These were not very informative, given the low power for these analyses.

Reviewer: 2

For a valid Mendelian randomization analysis, certain assumptions have to be fulfilled. The authors should describe in more details how these assumptions were assessed in the Methods, or if they were assessed previously, corresponding references should be provided. The absence of pleiotropy, one of these assumptions, is not depicted in Figure 1.

As discussed above, evidence for a robust association between genetic variant and smoking heaviness is well known in the literature. The pleiotropy assumption, as discussed in the paper, is assessed directly in our analysis by looking at associations between the variant and depression and anxiety in the never smokers. We have mentioned this in the methods section:

"The analysis in never smokers is a test of a key assumption of Mendelian randomisation: that the gene only operates on the outcome through its effects on smoking heaviness (i.e., no pleiotropy). If rs16969968/rs1051730 only operates on an outcome measure through smoking heaviness, no association should be observed in never smokers."

Mendelian randomization analysis can be implemented in two ways, 1) Using genetic polymorphisms as proxies of exposure; or 2) Instrumental variable analysis using genetic polymorphisms as instrument. There has been some discussion on the utility of these approaches recently (Methodological challenges in Mendelian randomization published in Epidemiology, 2014). Can the authors explain why they choose the first approach over the second approach in this study?

We have explained this in the discussion (page 18):

“For the same reason, we did not perform instrumental variable analysis to estimate the magnitude of the causal effect of smoking heaviness on depression or anxiety. It has been demonstrated that using cigarettes per day as an intermediate variable in Mendelian randomisation analyses using rs16969968/rs1051730 can lead to large biases in causal effect size estimates 49.”

The observational analysis is highly prone to confounding as only a few factors were adjusted for in this analysis. Therefore, even the result of a meta analysis is likely to be biased by methodological issues described in the Introduction. Can the authors adjust for more potential confounders such as sociodemographic factors in each of the study included?

We see little benefit in doing this as the observational estimates are still likely to be subject to residual confounding. We are using Mendelian randomisation to try to minimise confounding.

As the Mendelian randomization analysis only showed smoking heaviness (not smoking status) not associated with depression and anxiety among smokers, the conclusion in the abstract should be rephrased for better clarity.

We have rephrased the conclusion in the abstract.

Inconsistencies in referencing style in the reference list, and typos in the author (Davey Smith G instead of Smith GD)

This is due to automatic referencing software and will be best corrected in the final version of the document.